# A Functional 67-bp Duplication Locating at the Core Promoter Region within the Bovine *ADIPOQ* Gene Is Associated with Ovarian Traits and mRNA Expression

**DOI:** 10.3390/ani14162362

**Published:** 2024-08-15

**Authors:** Yufu Li, Tingting Liu, Mengyang Zhang, Chuanying Pan, Xu Liu, Haiyu Zhao, Xianyong Lan

**Affiliations:** 1College of Animal Science and Technology, Northwest A&F University, Yangling 712100, China; liyufu2022@126.com (Y.L.); liutingting1109@126.com (T.L.); zhangmengyang0518@163.com (M.Z.); chuanyingpan@126.com (C.P.); 2College of Veterinary Medicine, Northwest A&F University, Yangling 712100, China; liuxu1658@163.com; 3School of Life Sciences, Jiangsu Normal University, Xuzhou 221008, China; 4College of Life Science, Lanzhou University, Lanzhou 730000, China

**Keywords:** bovine, *ADIPOQ*, structural variation, ovarian, correlation analysis

## Abstract

**Simple Summary:**

Our previous investigation identified a 67-bp variable duplication in the *ADIPOQ* promoter region, which may impact cow ovarian development. Our researchers looked into the relationship between *ADIPOQ* promoter variants and ovarian traits and mRNA expression in Chinese Holstein cows in order to gain a better understanding of how to maximize the milk production of these cows while preserving their reproductive efficiency. Three genotypes were found in 2219 samples, and there was a strong correlation between this mutation and ovarian traits. Moreover, the mutant phenotype dramatically decreased the expression of the *ADIPOQ* gene mRNA in ovarian tissue. To increase dairy cow fertility, it is crucial to identify the genes that govern ovarian features, given the significance of the ovarian in the female reproductive system. In order to enhance the reproductive quality of Chinese Holstein dairy cows, this study offers a theoretical framework and molecular breeding signals.

**Abstract:**

*ADIPOQ* plays a crucial role in regulating the reproductive system, but there are few reports on the effects of *ADIPOQ* on ovarian in dairy cows. Previous studies have verified the presence of a 67-bp mutation in the promoter region of the *ADIPOQ* gene. Hence, we employed ovarian tissues *(n* = 2111) and blood samples (*n* = 108) from Chinese Holstein cows as experimental samples to examine the association between *ADIPOQ* promoter variants and ovarian traits. We extracted DNA from these samples and conducted genetic typing identification on each sample using advanced techniques like PCR and agarose gel electrophoresis. Consequently, the DD, ID, and II genotypes were discovered. and it has been observed that the mutation frequency of this locus is low in the Chinese Holstein cow. Importantly, the correlational analysis unveiled a significant relationship (*p* < 0.05) between the weight of ovaries in late estrus and the width of ovaries during the estrus interval with the mutation. Result of the RT-PCR revealed that the ID genotype partially diminished the expression of the *ADIPOQ* gene. The results of this study suggest that the identified variable duplication could serve as a potential genetic marker for enhancing the ovarian traits of Chinese Holstein cows.

## 1. Introduction

The Chinese Holstein cow, a breed developed through extensive crossbreeding and selection, stands as the only dairy breed in China and occupies a critical economic position due to its superior milk production traits. Furthermore, animal breeding is regarded as one of the most important economic sectors in a nation, with special importance [1]. Milk productivity is substantially increased after the application of various novel techniques for dairy breeding [2]. Similarly, breeders have primarily focused on milk production traits in Chinese Holstein cows. However, this emphasis has inadvertently led to a notable decline in the conception rate of cows, due to adverse genetic correlation between milk yield and production traits [3]. This is because progesterone and estradiol, two reproductive chemicals that create issues in the cow’s estrous cycle, are eliminated by the increased hepatic metabolism of high-producing cows [4]. Therefore, it is essential to prioritize research aimed at improving milk yield while maintaining reproductive efficiency. 

The ovaries are vital organ in female reproduction, playing a crucial role in the production of ova and the secretion of estrogen. The primary factors determining fertility in dairy cattle include follicle size, oocyte quality, timely development of the corpus luteum, and embryo quality. Fully developed, large follicles produce significant amounts of estradiol (E2), which is crucial for proper estrous behavior and timely insemination [5]. According to various studies, this primary organ for fertility has morphological variability. These variabilities are mainly associated with hormone levels and hereditary factors [6]. Thus, identifying essential genes regulating the activities of the ovaries, corpus luteum, and associated traits is fundamental for using molecular marker-assisted selection strategies to enhance cow fertility [7,8].

The multifunctional cytokine called adiponectin (*ADIPOQ*), also referred to as 30 kDa adipocyte complement-related protein (Acrp30), is secreted by adipose tissue and has a molecular weight of 30 kDa [9]. It was first discovered and cloned in 3T3-1 adipocytes by Scherer et al. in 1995 [10]. Since its discovery, *ADIPOQ* has been shown to increase insulin sensitivity, prevent atherosclerosis, inhibit inflammation, and participate in lipid metabolism [11]. This protein is predominantly expressed in adipose tissue and circulates in the blood, existing in various forms to perform diverse biological roles by binding to two receptors: adiponectin receptor 1 (Adipo R1) and adiponectin receptor 2 (Adipo R2) [12]. In mammalian liver, fat, and skeletal muscle, *ADIPOQ* regulates sugar and fat metabolism by affecting sugar utilization and insulin sensitivity. Consequently, most studies have focused on fat deposition in livestock. Numerous animal models have been developed to study the downstream effects of adiponectin signaling and function in diverse tissues in order to better understand the metabolic impacts of adiponectin. For instance, in sheep, mutations in the *ADIPOQ* gene have been found to correlate with growth traits and carcass traits, while in pigs these mutations have been associated with production and reproductive traits [13]. The bovine *ADIPOQ* gene is located on chromosome 1, near QTLs associated with marbling, eye muscle area, and dorsal marker thickness, and consists of five exons. Additionally, this gene has influence on weight traits and yield grade in cattle [14]. 

In 2005, the expression of *ADIPOQ* was identified in pig ovaries, demonstrating for the first time that *ADIPOQ* might play a role in the regulation of reproductive processes in animals [15]. Prior to this, *ADIPOQ* and its receptor had been detected in human ovaries, and *ADIPOQ* SNPs were associated with polycystic ovaries syndrome [16]. Some research has demonstrated that adipokines have an impact on female reproduction both directly and indirectly. Thus, it has been shown that *ADIPOQ* regulates the function of the gonads and the hypothalamic–pituitary axis [17]. Subsequent studies found that *ADIPOQ* was also expressed in rodents and livestock, but its level of expression varies among species [18]. *ADIPOQ* is now widely studied, and some progress has been made in studies in humans, rats, and pigs [14]. However, there are few reports about the effect of *ADIPOQ* on the ovarian tissue of cows. Since ovarian development is closely related to reproduction, it is crucial to investigate the factors that influence ovarian development in cows.

Our previous study reported a 67-bp variable duplication in the *ADIPOQ* promoter region that not only reduced the transcriptional activity of *ADIPOQ* in 3T3_L1 cells but also reduced its expression in adipose tissue [19]. In combination with the function of *ADIPOQ*, we speculate that it might affect ovarian development in cows. Therefore, the aim of the current study was to examine the correlation between *ADIPOQ* promoter polymorphism and ovarian development, which is expected to provide a theoretical basis for improving the fertility of Chinese Holstein cows.

## 2. Material and Methods 

### 2.1. Ethics Statement

The experimental protocols in this study have received approval from the Institutional Animal Care and Use Committee of the Northwest A&F University. The sample collection was carried out in compliance with the Chinese national standard “Guidelines for the Welfare and Ethical Review of Experimental Animals” (GB/T 35892-2018) [20]. 

### 2.2. Experimental Animal and Data Collection

Data and samples were gathered from 2219 healthy adult Chinese Holstein cows (4–5 years old) from well-established dairy farms in Shaanxi Province, China (Figure 1), all of whom received the same nutrition and feeding guidelines and were managed according to the Regulations on Livestock and Poultry Identification and Farming Record Management [21]. The whole-blood samples were collected from the tail veins of 108 cows. The ovaries of the remaining 2111 cows were taken right after they were euthanized. Subsequently, the tissues were put in clean sterile saline solution and sent to the laboratory within two hours. Once the connective and fatty tissues were cut away from the surface of ovaries, measurements were taken of their length, width, height and weight, number and size of mature follicles, number and size of corpus luteum, and number and size of corpus albicans. Only healthy Holstein cows in the same age phase and physiological stage were included in our sample, since follicle size and development, and therefore the dimensions of the ovary, are dependent on the stages of the estrous cycle. Moreover, we identified the estrus stage indirectly by typing the corpus luteum: types 1 and 2 indicated the late stages of estrus; types 3 and 4 indicated the dioestrum; the lack of follicles and corpus luteum on the ovaries indicated pre-estrus; and the absence of corpus luteum and large follicles indicated estrus (types of corpus luteum: conical, type 2: volcanic crater, type 3: mushroom-shaped, type 4: flattened, and type 5: no corpus luteum). After the collection of phenotypic data, the ovarian tissues were rinsed three times with PBS and quickly frozen in a refrigerator set at −80 °C for subsequent experiments [22].

### 2.3. DNA Extraction from Different Tissue

The phenol–chloroform method was employed to extract genomic DNA from blood samples and ovarian tissue samples, in accordance with to previous reports [22]. The DNA samples were evaluated for quality by determining the A260/A280 ratio using a NanoDrop 1000 spectrophotometer [23]. The DNA samples were diluted to 50 ng/µL and stored in a refrigerator at −40 °C for subsequent experiments.

### 2.4. PCR Amplification and Sequence Analysis

The primer sequences, which were in accordance with the findings of earlier investigations (Table 1), were synthesized by Shanghai Bioengineering Biological Co. (Shanghai, China) [19]. The PCR amplification apparatus and protocol followed the methods described by Liu et al. [22]. A volume of 5 μL of PCR product was subjected to electrophoresis on a 3.0% agarose gel at a voltage of 120 V for 40 min (Figure 1). Subsequently, images were captured using a gel imaging system. If the product fragment size met the expected criteria, PCR products of various genotypes were purified by the SanPrep Column PCR Product Purification Kit. The refined product was ligated into a T vector (pMD19-T) and transformed into DH5α-sensitive cells. Subsequently, the recombinant plasmid was amplified using PCR and sent to Xi’an TsingKe Jersey Biotechnology Co. (Xi’an, China) for sequencing [24]. 

### 2.5. Molecular Evolutionary Tree Construction 

An initial alignment was conducted using ClustalW comparison with default parameters, using the protein sequences of adiponectin from bovine, sheep, goat, pig, horse, dog, Norway rat, chimpanzee, and human sources from the NCBI database. MEGA11 software was then used to generate evolutionary trees [25]. The neighbor-joining method was implemented to construct the tree, with a *p*-distance, bootstrap procedure with 1000 duplicates, partial deletion, and a 50% site coverage cutoff. The resulting phylogenetic tree was visualized using the Interactive Tree Of Life online tool (https://itol.embl.de/, accessed on 23 July 2024).

### 2.6. cDNA Synthesis and Quantitative RT-PCR

To identify the effects of different genotypes on ADIPOQ mRNA expression level, a total of six Chinese Holstein cows were randomly selected for RT-PCR analysis, based on the genotyping results (DD = 3, ID = 3). Each sample was subjected to three technical replicates, to ensure robustness in the results. Total RNA was extracted from ovarian tissue using TRIzol total RNA extraction reagent (TaKaRa Biotechnology Co. Ltd., Dalian, China) and stored at −80 °C. According to the manufacturer’s protocol, the First-strand cDNA was synthesized using the PrimeScript TM RT kit (TaKaRa Biotech Co. Ltd.), and then all genome cDNA was stored at −20 °C. According to the *ADIPOQ* gene sequence in GeneBank, a pair of quantitative polymerase chain reaction (qPCR) primers were designed, as shown in Table 1. A CFX96 real-time PCR detection system (Bio-Rad, Hercules, CA, USA) was used for qPCR; the reaction contained 5 µL SYBR Premix Ex TaqTMII (TaKaRa Biotech Co. Ltd.) and the upstream and downstream primers were each 0.5 µL, cDNA template 1 µL, ddH_2_O 3 µL. The reaction conditions are as follows: initial denaturation at 95 °C for 3 min, followed by 40 cycles of 95 °C for 10 s and 55 °C for 30 s [26]. qPCR analysis of cDNA derived from each tissue was performed in triplicate, and relative gene expression was normalized using that of β-actin by using the 2^−ΔΔ^Ct method, as described previously [27].

### 2.7. Statistical Analyses

Data on ovarian traits were tallied using the Excel program. Population genotype frequency, allele frequency, and the Hardy–Weinberg equilibrium test (Hardy–Weinberg equilibrium, HWE) were calculated by chi-squared test. The classification results were calculated using Excel. The degree of purity (homozygosity, Ho), heterozygosity (heterozygosity, He), the number of effective (effective number of alleles, Ne), and the polymorphism information content (polymorphism information content, PIC) were calculated using the online software Genetic Diversity Index Calculator (http://www.msrcall.com/Gdicall.aspx, accessed on 15 June 2024) [28]. Correlation analysis of genotypes and ovarian traits was calculated by SPSS 27.0 software (SPSS, Inc., Chicago, IL, USA), and the statistical model is as follows:$$Y_{\mathrm{ijk}}=\mu \;+\;G_{i}+e_{\mathrm{ijk}}$$

Here, Y_ijk_ stands for the phenotypic value of each ovarian trait; µ represents the overall population mean; G_i_ stands for the fixed effect of genotype; and e_ijk_ stands for random error [29]. In this model, the impacts of age, sex, sampling season, and rearing environment were not taken into account because they were constant. We utilized the least squares mean with standard deviation for various genotypes and ovarian traits. All data were expressed as mean ± standard error, and *p* < 0.05 was considered statistically significant [22]. In the meantime, the nonparametric (Kruskal–Wallis) test in SPSS 27.0 software was used to assess the data that did not follow normal distribution and homogeneity of variances [30]. 

## 3. Results

### 3.1. Results of the Variable Duplication Locus Genotyping in the ADIPOQ

The PCR products were confirmed using agarose gel electrophoresis, and the outcomes are depicted in Figure 2A. The electrophoresis results indicated the presence of *ADIPOQ* genotypes in the Chinese Holstein cow population, specifically DD (originally named 1D/1D) for the homozygote reference with a product fragment size of 471 bp, ID for the heterozygous genotype (originally named 1D/2D), and II for the homozygote variant genotype (originally named 2D/2D) with two product fragments of 538 bp. These findings align with our anticipated results and provide a basis for conducting additional experiments.

### 3.2. ADIPOQ Gene Sequence and Molecular Evolution Analysis 

PCR results from both genotypes were chosen at random for sequencing confirmation, as depicted in Figure 2B,C. Sequence analysis identified a 67-base pair repetitive segment in both DD and ID genotypes, as well as two repetitive fragments in II genotypes, which aligns with earlier research. As can be seen in Figure 2D, the evolutionary relationships between ADIPOQ genes and species was largely consistent with traditional classification, with cattle showing high affinity with sheep and goats as the closest homologues.

### 3.3. Estimation of Genetic Parameters of the 67-bp Mutation in the Chinese Holstein Cow Population 

For the *ADIPOQ* promoter region, Table 2 shows the population genetic characterization of the variable repeat sequences. Three different genetic compositions were identified in the samples of ovarian tissue. The frequency of the DD genotype was 0.986, the frequency of the ID genotype was 0.013, and the frequency of the II genotype was 0.0005. The frequency of the “D” allele was 0.993, which was higher than that of the “I” allele (0.007), indicating that the “D” allele was the dominant allele in the population. The polymorphic information content (PIC) value for this locus was 0.014, indicating low polymorphism (PIC < 0.25). In addition, the χ^2^ test revealed a deviation from Hardy–Weinberg equilibrium for this polymorphic locus (*p* < 0.05). However, cows with blood samples exhibited only the DD genotype.

### 3.4. Correlation Analysis of ADIPOQ Gene Variants with Ovarian Traits

Researchers investigated the relationship between different stages of reproduction and duplication variation in the promoter region of the *ADIPOQ* gene in Chinese Holstein cows. However, this correlation was not observed during pre-estrus and estrus (Appendix A). Yet, according to the data presented in Table 3, there was a significant correlation (*p* < 0.05) between the weight of the ovaries during late estrus and the width of the ovaries during the estrus interval, with the duplication variation in the promoter region of the *ADIPOQ* gene. Furthermore, individuals with the DD type had significantly higher weight and greater width of the ovaries compared to individuals with the ID type. Nevertheless, there was no notable link observed for mature follicles, corpus luteum, or corpus albicans.

### 3.5. ADIPOQ mRNA Expression Analyses in Different Genotypes 

Based on the genotyping results, six ovarian tissues were randomly selected for tissue RNA extraction and reverse transcription to cDNA. qRT-PCR was used to identify the expression of three different genotypes of the *ADIPOQ* gene in ovarian tissues. The results revealed that the expression levels of ID genotypes were significantly lower than those of the DD genotype (Figure 3). Considering that only one individual had genotype II, we examined the mRNA expression trend of the *ADIPOQ* gene in this individual and found that its expression was close to that of the ID genotype.

## 4. Discussion

In Chinese Holstein cow production practices, there is a tendency to over-enhance production traits in order to increase milk production, which has resulted in a significant decline in herd reproductive performance, posing a significant challenge to the long-term development of the Chinese Holstein cow industry [31]. The herd’s reproductive performance is an important driver of production levels, and most reproductive factors exhibit poor heritability. The ovaries are vital for animal reproduction because they produce oocytes and secrete estrogen. Reproductive hormone fluctuations in cows are strongly correlated with the developing state of the ovaries during the estrous cycle. Reproductive success and the stability of subsequent pregnancies in cows are impacted by changes in the size and weight of the ovaries during the estrous cycle, which also directly affects the amount and quality of oocytes [32]. Therefore, research on marker genes linked to ovarian features advances the dairy business by increasing the reproductive efficiency of cows.

Previous studies have shown that the adiponectin system, consisting of adiponectin, its receptors AdipoR1 and AdipoR2, and the APPL1 complex, is found in both the bovine ovaries and embryos. Adiponectin interacts with its receptors and exerts an influence on ovarian function in cattle [33]. The promoter is a component of the gene that acts like a “switch”, which determines gene activity, and mutations in the promoter part of the gene can lead to dysregulation of gene expression [34]. Our previous results have confirmed that a 67-bp variable duplication in the promoter region of the *ADIPOQ* gene could affect transcriptional activity and the expression of *ADIPOQ* mRNA [19]. Genetic polymorphism analysis is a measure of the magnitude of genetic variation in a population [35]. Previous testing conducted on Xinjiang Brown cattle revealed the presence of only two genotypes [22]. Our findings revealed the presence of three genotypes—II, DD, and ID—in the Chinese Holstein cow population at this specific locus. This difference might be attributed to different geographical locations and breeding environments. The prevalence of allele “D” in the Chinese Holstein cow population at a relatively high frequency suggests that “D” is the dominant allele. The “DD” genotype was determined to be the prevailing genotype and can serve as a molecular marker for future investigations. This may be the result of natural selection, with the II genotype being gradually eliminated.

Furthermore, research has demonstrated a substantial correlation between *ADIPOQ* polymorphisms and the risk of developing polycystic ovarian syndrome in people [36]. In this study, we examined the correlation between genetic variants in the core promoter region of the *ADIPOQ* gene and traits related to bovine ovaries. The weight of the ovary in late estrus (*p* = 0.012) and the width of the ovary in the estrus interval (*p* = 0.028) were shown to be significantly correlated with the mutation site of the *ADIPOQ* gene. Considering the effect of different genotypes on the expression of the *ADIPOQ* gene, we examined the expression level of the *ADIPOQ* gene in ovarian tissues of the three genotypes by RT-qPCR, and the results were consistent with expectations. Both homozygote (II) and heterozygotes (ID) suppressed the expression of the *ADIPOQ* gene. However, a small number of mutant heterozygotes still exist, and these affect the development of bovine embryos by suppressing the expression of the *ADIPOQ* gene. In summary, the DD genotype can be used as a molecular marker to screen cows for superior traits and further improve herd fertility.

As the most abundantly expressed protein secreted by adipocytes, the *ADIPOQ* gene has various biological functions, and its role in animal reproduction has been a hotspot in recent studies [35]. The correlation between the *ADIPOQ* gene promoter polymorphism and ovarian traits was analyzed for the first time in a large sample size of Chinese Holstein cow ovarian tissues in this experiment. The findings verified that the polymorphic locus exhibited a highly significant correlation with ovarian weight and width; however, additional research is required to clarify its specific regulatory mechanism.

## 5. Conclusions

This study investigated the relationship between ovarian traits and the 67-bp variable duplication in the promoter region of the *ADIPOQ* gene in Chinese Holstein cows. The DNA from samples of blood and ovarian tissue was extracted and amplified using PCR. Agarose gel electrophoresis was then used to determine the polymorphisms present in the samples. Three genotypes were found to be present in these samples: II, DD, and ID. The DD genotype exhibited a greater frequency compared to the ID and II genotypes. We found that there is a significant relationship between this variant and the width of the estrus-interval ovaries and the weight of the late-estrus ovaries, through the use of correlation analysis. Additionally, we discovered, by using qPCR assays, that the mutant phenotype dramatically reduced the *ADIPOQ* gene mRNA expression in ovarian tissue. Consequently, our discoveries can serve as a molecular breeding indicator for Chinese Holstein cows and establish a theoretical basis for enhancing their reproductive quality.

## Figures and Tables

**Figure 1 animals-14-02362-f001:**
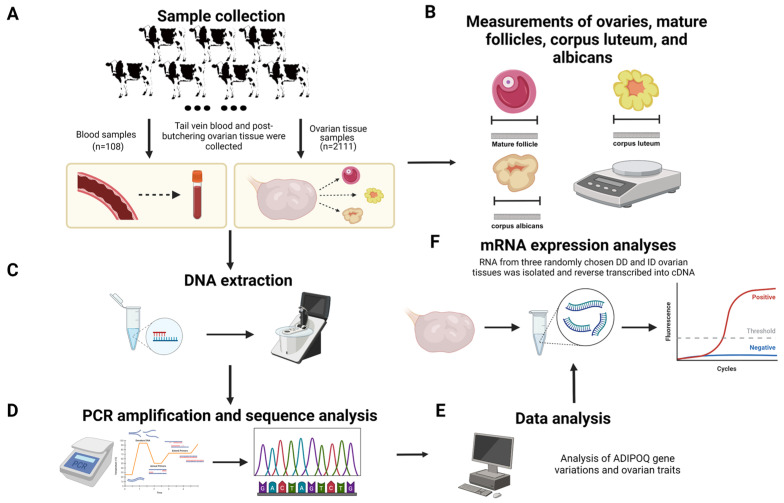
Research strategies for examining *ADIPOQ* promoter polymorphisms and ovarian development (created with BioRender.com, accessed on 23 July 2024). (**A**) Sample collection. (**B**) Ovarian traits data measurement. (**C**) DNA extraction. (**D**) PCR amplification and sequencing. (**E**) Data analysis. (**F**) mRNA extraction and RT-PCR.

**Figure 2 animals-14-02362-f002:**
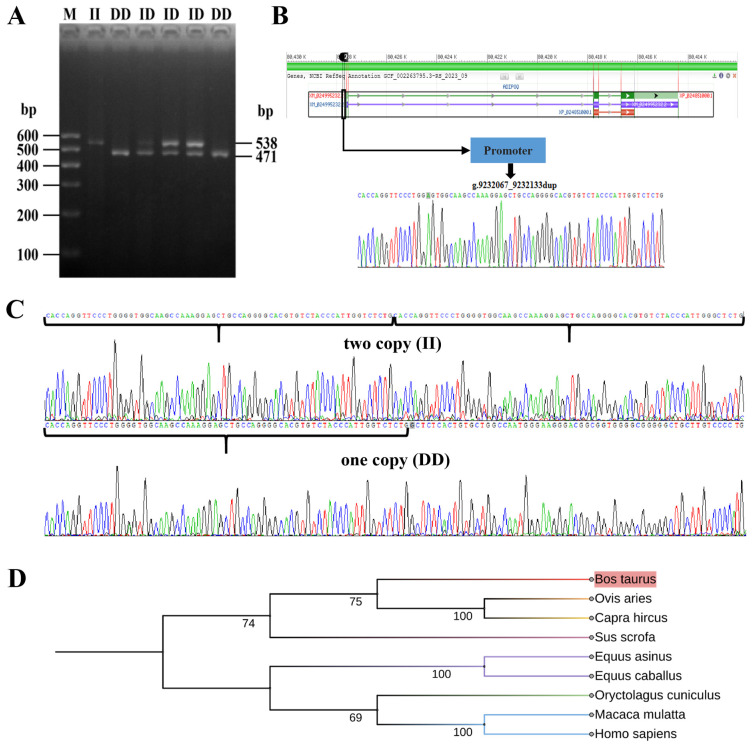
Identification and analysis of 67-bp duplication in the promoter region of the *ADIPOQ* gene. (**A**) The electrophoresis diagrams of the PCR amplification products. DD (471 bp): representing the wild type, contains one copy; ID (471 and 538 bp): defines the heterozygote. (**B**) Schematic representation of *ADIPOQ* gene polymorphic loci. (**C**) The sequence of the *67-bp* duplication. II genotype above, DD genotype below. (**D**) Molecular evolutionary tree of the *ADIPOQ* gene.

**Figure 3 animals-14-02362-f003:**
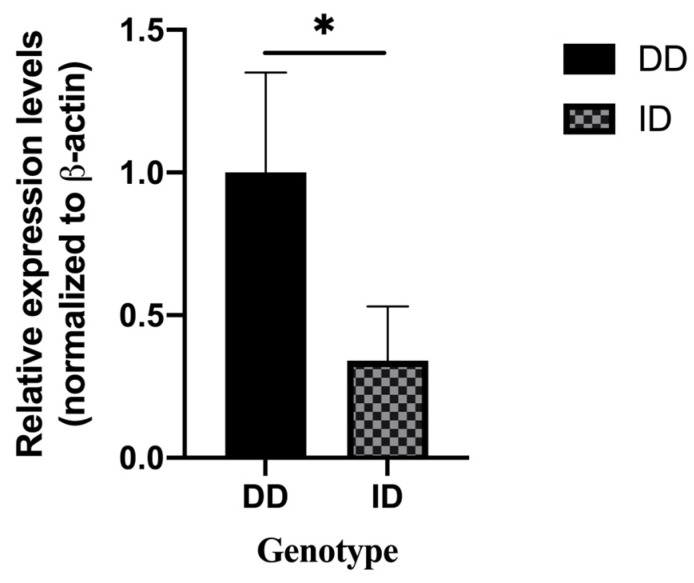
mRNA expression of different genotypes of the cow *ADIPOQ* gene in ovarian tissue (DD = 3, ID = 3), “*” means: *p* < 0.05.

**Table 1 animals-14-02362-t001:** Primer information for the *ADIPOQ* gene in cattle.

Primer	Sequence (5′-3′)	Length	Function	References
*ADIPOQ*-F1	AGAAATGTTCCCTCACCTCAGT	471	PCR	[19]
*ADIPOQ*-R1	CTCGGTACTCATGGGGAC			
*ADIPOQ*-F2	ATTCCACACCTGAGGGGCT	92	qRT-PCR	-
*ADIPOQ*-R2	TCTTCCATGTTGTCCTCGCC			
β-actin-F1	CAAGGCCAACCGTGAGAA	96	Reference gene	-
β-actin-R1	GCATACAGGGACAGCACAG			

**Table 2 animals-14-02362-t002:** Population genetic parameters of the *ADIPOQ* gene in ovarian tissue samples.

Sample	Sample Size	Genotypic Frequencies				Allelic Frequencies		HWE	Population Parameters			
		II	ID	DD	*p* Values	D	I	*p* Values	Ho	He	Ne	PIC
Ovarian Tissue	2111	0.0005	0.013	0.9865	*p* < 0.05	0.993	0.007	*p* < 0.05	0.986	0.014	1.015	0.014

Note: HWE, Hardy–Weinberg equilibrium. Ho, observed homozygosity. He, heterozygosity. Ne, effective allele numbers. PIC, polymorphism information content.

**Table 3 animals-14-02362-t003:** Relationship between the 67-bp duplication variation within the *ADIPOQ* gene and the ovarian traits (late estrus and estrus interval) of Chinese Holstein (LSM ± SE).

Oestrus Cycle	Quantitative Traits		Genotypes		*p*-Value
			DD	ID	
Late estrus	Ovary	length (mm)	42.6 ± 0.4 (*n* = 380)	36.8 ± 1.9 (*n* = 8)	0.059
		width (mm)	22.1 ± 1.4 (*n* = 380)	22.3 ± 2.7 (*n* = 8)	0.957
		height (mm)	28.5 ± 0.3 (*n* = 379)	26.1 ± 1.8 (*n* = 8)	0.306
		weight (g)	12.5 ± 0.3 (*n* = 380)	8.0 ± 0.5 (*n* = 8)	**0.012 ***
	Mature follicle	number	0.7 ± 0.1 (*n* = 246)	0.8 ± 0.4 (*n* = 5)	0.843
		diameter (mm)	8.5 ± 0.5 (*n* = 194)	11.0 ± 2.1 (*n* = 3)	0.538
	Corpus luteum	number	1.2 ± 0.0 (*n* = 380)	1.0 ± 0.0 (*n* = 8)	0.279
		diameter (mm)	18.3 ± 0.4 (*n* = 373)	17.4 ± 2.1 (*n* = 8)	0.724
Estrus interval	Ovary	length (mm)	42.1 ± 0.3 (*n* = 1083)	40.9 ± 2.5 (*n* = 11)	0.642
		width (mm)	21.8 ± 0.2 (*n* = 1083)	19.4 ± 0.9 (*n* = 11)	**0.028 ***
		height (mm)	24.9 ± 0.2 (*n* = 1083)	25.5 ± 3.0 (*n* = 11)	0.763
		weight (g)	11.2 ± 0.2 (*n* = 1082)	9.2 ± 1.4 (*n* = 11)	0.212
	Mature follicle	number	0.4 ± 0.0 (*n* = 919)	0.4 ± 0.3 (*n* = 7)	0.916
		diameter (mm)	4.9 ± 0.2 (*n* = 702)	4.4 ± 3.2 (*n* = 4)	0.874
	Corpus luteum	number	1.7 ± 0.0 (*n* = 1085)	1.5 ± 0.3 (*n* = 11)	0.531
		diameter (mm)	11.6 ± 0.3 (*n* = 903)	12.8 ± 2.4 (*n* = 11)	0.609
	Corpus albicans	number	0.6 ± 0.0 (*n* = 708)	0.3 ± 0.3 (*n* = 3)	0.728
		diameter (mm)	1.5 ± 0.1 (*n* = 680)	1.3 ± 1.3 (*n* = 3)	0.894

Note: “*” means: *p* < 0.05. The bold font indicates a significant difference. Genotypes are omitted if the number of individuals is fewer than 3.

## Data Availability

Data are contained within the article.

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
