# Peer review of "A Functional 67-bp Duplication Locating at the Core Promoter Region within the Bovine ADIPOQ Gene Is Associated with Ovarian Traits and mRNA Expression"

_animals, 2024, doi:10.3390/ani14162362_

Round 1

Reviewer 1 Report

Comments and Suggestions for Authors

The manuscript needs revision. Please refer to comments given in the text of reviewed attached file of the manuscript.

Author Response

[Comments 1] It is better to add in abstract shortly about DNA extraction, PCR, ....

[Response 1] Thank you for advice. We have made the necessary additions (SEE the revised first page).

[Comments 2] The conclusion is very wide and general. It is better to add a specific conclusion from your specific results

[Response 2] Thank you for suggestion. We have made the necessary additions(SEE the first page of the revised version).

[Comments 3] Please use space before [ through out all text of the manuscript.

[Response 3] Thank you for suggestion. We have made the necessary modifications as per your suggestions. We appreciate your thorough review and guidance on this matter.

[Comments 4] What is the superiority of your research compared to other researches?

[Response 4] Thank you for your valuable feedback. We appreciate your interest in our research. Our study focuses on investigating the regulation of the ADIPOQ gene on ovarian traits in dairy cows, which is crucial for understanding ovarian potential and assessing female reproductive capacity. Recognizing the significance of ovarian weight in reflecting the potential for follicle production, we have utilized ovarian weight as an effective indicator for estimating female reproductive ability. While the ovaries play a vital role in female fertility, there is a scarcity of literature on excellent genes that control ovarian morphological features. Our research aims to address this gap by investigating the ADIPOQ gene with a sufficient sample size, providing new theoretical foundations for the breeding of Chinese Holstein dairy cows. Most studies on the ADIPOQ gene currently focus on adipose traits, with limited research on its regulation in dairy cow reproduction. Our work seeks to fill this void, offering a fresh perspective and contribution to this field. We believe that our study, through in-depth exploration of the relationship between the ADIPOQ gene and ovarian traits, holds a unique advantage in unraveling the reproductive regulatory mechanisms in dairy cows.

[Comments5] this figure belongs to materials and methods section, please move to there.

[Response 5] We are grateful for your time. The aforementioned figure will be moved as soon as possible from its present location to the Materials and Methods section per your recommendation.

[Comments 6] please add country

[Response 6] Thank you for your valuable suggestions. We agree with this comment and have made the necessary additions. You can see these on the third page of the revised version.

[Comments 7] Please add reference for used DNA extraction method.

[Response 2] Thank you for your insightful comments. We agree with this comment and have made the necessary additions. You can see these on the fourth page of the revised version.

[Comments 8] please identify in the text, did you have biological repeats and technical repeats? and how many? for RT-PCR?

[Response 8] Thank you very much for the question. We have biological replicates as well as technical replicates. In order to clarify the effects of different genotypes on ADIPOQ mRNA expression levels, 6 Chinese Holstein cows were randomly selected for RT-PCR based on the genotyping results(DD=3, ID=3), and each sample had three technical replicates .We have made the additions on the fourth page of the revised version.

[Comments 9]Please identify in the text. please add reference for used softwares and programs!

[Response 9] We appreciate your suggestion to include references for the software and programs utilized in our study. In the revised manuscript, we will ensure to explicitly identify and provide appropriate references for all software and programs used in our research.

[Comments 10] did you use animal statistical model? Please add this model and its components in the text

[Response 10] Thank you very much for the question. To address your question on the statistical model we used in our investigation, we did use an animal model. To be transparent and clear about our process, we will include a thorough explanation of this model and all of its parts in the updated text. We have made the additions on the fifth page of the revised version.

[Comments 11] The conclusion is very wide and general. It is better to add a specific conclusion from your specific results

[Response 11] Thank you for your valuable feedback on the breadth of our conclusion. We acknowledge the need to refine our conclusion by incorporating specific findings from our study. We will enhance the specificity and relevance of our conclusion by highlighting specific conclusions drawn from our results. You can see the modifications in the conclusion section on page nine.

Reviewer 2 Report

Comments and Suggestions for Authors

This research is about “the correlation between ADIPOQ promoter polymorphism and ovarian development”. This research provides interesting information. However, some important changes are needed before final publication.

Simple abstract: I recommend restructuring this section because it is the same as the “abstract”. In the “simple abstract” the main results should be explained in an informative way focusing on the repercussions in the bovine sector.

INTRODUCTION

General comments:

I recommend expanding this section on the information available on “Adiponectin (ADIPOQ)”, functions, where is it secreted?. Also, be clearer in its justification and hypothesis.

Specific comments:

Lines 44-46.- mention “However, this emphasis has inadvertently led to a notable decrease in the conception rate of cows, due to the adverse genetic correlation between milk production and production traits[2]”. I recommend mentioning a reference that mentions that this is because high-producing cows have a higher hepatic metabolism that eliminates reproductive hormones such as progesterone and estradiol, which causes problems in the cow's estrous cycle.

Line 91.- change “wasto” to “was to”

Line 92.- They mention “Figure 1”, I recommend changing it because this part should be described in the “Material and Methods” section

MATERIAL AND METHODS

Specific comments:

Line 100-102.- they mention “The care and usage of animals in this experiment strictly adhere to local policy standards and animal welfare legislation”. Which ones? I recommend adding the author and year.

Line 104-105.- they mention “Data were collected from Chinese Holstein cows. A total of 2219 healthy adult cows were selected”. I recommend mentioning the selection criteria, exclusion, age of the cows, lactations, management, etc.

Line 106.- they mention “nutritional conditions were the same”, how much? “mean±SE”, mention the body condition scale.

Line 107.- they mention “The ovaries of the remaining 2111 cows were taken right after they were euthanized”. Did you control the “euthanized”?

Line 163.- in the “Statistical analyses” section. I recommend specifying the nature of the data, whether they came from a normal distribution, were they transformed? Were there outlier data?

Results General comments: Where in this section is “fig 2(B)Schematic representation of ADIPOQ gene polymorphic loci” mentioned?

Particular comments: Line 192-198.- mention “Sequence analysis identified a 67-base pair repetitive segment in both DD and ID genotypes, as well as two repetitive fragments in II genotypes, which aligns with earlier research. This finding validates the presence of a polymorphic mutation in the ADIPOQ gene within the experimental population, indicating the need for further investigation. As can be seen in Figure 2D, the evolutionary relationships between ADIPOQ genes and species was largely consistent with traditional classification, with cattle showing high affinity to sheep and goats as the closest homologues.” I recommend this part be changed to the “Discussion” section.

Line 236.- Table 3, change “late estrus” to “Late estrus”. “estrus interval” for “Estrus interval” Line 249.- Fig 3, adjust figure format

DISCUSSION

In general, I recommend restructuring this section. Add more on “correlation between genetic variants in the core promoter region of the ADIPOQ gene and traits related to bovine ovaries”. Also where the findings in table 3 are mentioned

-Late estrus/“weight (g); (DD)12.5±0.3(n=380); (ID)8.0±0.5(n=8); (P-value)0.012*”

-Estrus interval/“width (mm); (DD)21.8±0.2(n=1083); (ID)19.4±0.9(n=11); (P-value)0.028*”

Particular comments:

Line 279.- Change “adiponectin” to “Adiponectin”

Line 288-293.- mention “Both homozygote (II) and heterozygotes (ID) suppressed the expression of ADIPOQ gene. This may be the result of natural selection with the II genotype being gradually eliminated. However, a small number of mutant heterozygotes still exist, which affect the development of bovine embryos by suppressing the expression of the ADIPOQ gene. In summary, the DD genotype can be used as a molecular marker to screen cows for superior traits and further improve herd fertility.” I recommend homogenizing with what was mentioned in line 273-276.

Conclusion

I recommend restructuring this section and being more specific about the findings mentioned in the results and described in “Discussion”.

Line 303-304.- I recommend eliminating “The samples used were ovarian tissue (n=2111) and blood (n=108)”.

Author Response

[Comments 1] Simple abstract: I recommend restructuring this section because it is the same as the “abstract”. In the “simple abstract” the main results should be explained in an informative way focusing on the repercussions in the bovine sector.

[Response 1] Thank you for your guidance and suggestion to reorganize this section to distinguish it from the abstract. In response, we will refine this segment to offer a simplified summary that accentuates the key findings in an informative manner, underscoring their significance within the cattle industry. Your advice will assist us in enhancing the clarity and impact of our research presentations.

[Comments 2] INTRODUCTION General comments: I recommend expanding this section on the information available on “Adiponectin (ADIPOQ)”, functions, where is it secreted?. Also, be clearer in its justification and hypothesis.

[Response 2] Thank you for your valuable guidance and suggestion. We acknowledge the importance of expanding the section concerning "Adiponectin (ADIPOQ)" by providing detailed information on its functions and secretion sites. In response to your suggestion, we will enrich this segment to offer a more comprehensive overview of ADIPOQ, including its functions and secretion locations. Additionally, we will enhance the clarity of our justification and hypothesis to provide a more robust foundation for our research. Your insights will help us improve the depth and clarity of our work in this area.

[Comments 3] Lines 44-46.- mention “However, this emphasis has inadvertently led to a notable decrease in the conception rate of cows, due to the adverse genetic correlation between milk production and production traits[2]”. I recommend mentioning a reference that mentions that this is because high-producing cows have a higher hepatic metabolism that eliminates reproductive hormones such as progesterone and estradiol, which causes problems in the cow's estrous cycle.

[Response 3] Thank you for your insightful recommendation. In line with your suggestion, we will include the following statement : "However, this emphasis has inadvertently led to a notable decline in the conception rate of cows, due to adverse genetic correlation between milk yield and production traits[3]. This is because progesterone and estradiol, two reproductive chemicals that create issues in the cow's estrous cycle, are eliminated by the increased hepatic metab-olism of high-producing cows[4]" Incorporating this information will provide a more comprehensive understanding of the challenges faced due to the genetic correlations between milk production and reproductive traits. Your input will enrich the context and depth of our research.

[Comments 4] Line 91.- change “wasto” to “was to”

[Response 4] Thank you very much for pointing out the will, which we have corrected. You can see these on the first page of the revised version.

[Comments 5] Line 92.- They mention “Figure 1”, I recommend changing it because this part should be described in the “Material and Methods” section

[Response 5] We are grateful for your input. The aforementioned figure will be moved as soon as possible from its present location to the Materials and Methods section per your recommendation.

MATERIAL AND METHODS

[Comments 6] Line 100-102.- they mention “The care and usage of animals in this experiment strictly adhere to local policy standards and animal welfare legislation”. Which ones? I recommend adding the author and year.

[Response 6] Thanks to your suggestions, we have revised this section to include specific details about local policy standards and animal welfare legislation that were adhered to during experiments. Your suggestions will help improve the clarity and credibility of our research regarding the care and treatment of animals according to established regulations.

[Comments 7] Line 104-105.- they mention “Data were collected from Chinese Holstein cows. A total of 2219 healthy adult cows were selected”. I recommend mentioning the selection criteria, exclusion, age of the cows, lactations, management, etc.

[Response 7] Thank you for your insightful feedback on the selection criteria and details of the study population. We appreciate your attention to these important aspects of our study. In response to your suggestions, we acknowledge the need for a fuller description of the selection criteria and characteristics of the study population. We did collect data from Chinese Holstein cows, with a total of 2219 healthy adult cows participating in the study. In order to provide you with a clearer understanding of our methodology, we have added more detailed information on the selection criteria, and the management methods used during the study. We are committed to addressing your concerns and will ensure that the subjects are described in detail in the revised manuscript to meet the standards of scientific rigour. Thank you for your valuable comments and guidance.

[Comments 8] Line 106.- they mention “nutritional conditions were the same”, how much? “mean±SE”, mention the body condition scale.

[Response 8] Thank you for your interest in the specific details of our study regarding the Nutritional Status and Physical Condition Scale. Your feedback is invaluable in ensuring the clarity and integrity of our results. The cows we selected were all healthy, and the level of husbandry management was in accordance with the Livestock Labelling and Breeding Records Management Scheme (2006). We have added this section in the revised manuscript. Thank you for your thorough review and constructive feedback. Your insights will undoubtedly help to improve the overall quality and clarity of our study.

[Comments 9] Line 107.- they mention “The ovaries of the remaining 2111 cows were taken right after they were euthanized”. Did you control the “euthanized”?

[Response 9] Thank you for highlighting the mention regarding the ovaries being collected from the remaining 2111 cows after euthanasia. We would like to clarify that the process of euthanasia was conducted in accordance with approved ethical guidelines and regulations. The welfare and ethical treatment of the animals involved in the study were of utmost importance to us.

[Comments 10] Line 163.- in the “Statistical analyses” section. I recommend specifying the nature of the data, whether they came from a normal distribution, were they transformed? Were there outlier data?

[Response 10] We appreciate your suggestion to provide more details about the nature of the data used in our analysis. In response, we will include information about whether the data originated from a normal distribution, if any transformations were applied, and if there were outlier data points that were addressed in the analysis. These additions will offer a clearer understanding of the data processing methods employed in our study. Thank you for highlighting these important aspects for clarification.

[Comments 11] Results General comments: Where in this section is “fig 2(B)Schematic representation of ADIPOQ gene polymorphic loci” mentioned?

[Response 11] Thank you for your observation of "Figure 2(B) Schematic diagram of polymorphic sites in the ADIPOQ gene" mentioned in the Results section. We have now added a reference to Figure 2(B) in Results 3.2 in the text to address this issue. Thank you very much for your attention to detail and we have made the necessary adjustments to ensure the clarity and coherence of our manuscript.

[Comments 12] Particular comments: Line 192-198.- mention “Sequence analysis identified a 67-base pair repetitive segment in both DD and ID genotypes, as well as two repetitive fragments in II genotypes, which aligns with earlier research. This finding validates the presence of a polymorphic mutation in the ADIPOQ gene within the experimental population, indicating the need for further investigation. As can be seen in Figure 2D, the evolutionary relationships between ADIPOQ genes and species was largely consistent with traditional classification, with cattle showing high affinity to sheep and goats as the closest homologues.” I recommend this part be changed to the “Discussion” section.

[Response 12]Thank you for your feedback regarding lines 192-198 of the manuscript. I have removed the redundant content from that section. Your suggestion to move this information to the "Discussion" section has been duly noted and implemented. Your input is valuable in ensuring the coherence and effectiveness of our manuscript. Thank you for your attention to detail.

[Comments 13] Line 236.- Table 3, change “late estrus” to “Late estrus”. “estrus interval” for “Estrus interval” Line 249.- Fig 3, adjust figure format

[Response 13] Thank you for your attention to detail and your suggestions for improving the presentation of our findings. We have made the necessary adjustments based on your suggestions.

DISCUSSION

[Comments 14] In general, I recommend restructuring this section. Add more on “correlation between genetic variants in the core promoter region of the ADIPOQ gene and traits related to bovine ovaries”. Also where the findings in table 3 are mentioned

-Late estrus/“weight (g); (DD)12.5±0.3(n=380); (ID)8.0±0.5(n=8); (P-value)0.012*”

-Estrus interval/“width (mm); (DD)21.8±0.2(n=1083); (ID)19.4±0.9(n=11); (P-value)0.028*”

[Response 14] Thank you for your guidance and advice. I have reorganized this section as suggested and included more details on the correlation between genetic variation in the core promoter region of the ADIPOQ gene and bovine ovary-related traits. The findings in Table 3 regarding late estrus and interestrus are now explicitly mentioned in the revised section. Your guidance helps improve the clarity and depth of our discussions in this area.

[Comments 15] Line 279.- Change “adiponectin” to “Adiponectin”

[Response 15] Thank you for bringing this to my attention. I have made the requested change. "adiponectin" has been updated to "Adiponectin".

[Comments 16] Line 288-293.- mention “Both homozygote (II) and heterozygotes (ID) suppressed the expression of ADIPOQ gene. This may be the result of natural selection with the II genotype being gradually eliminated. However, a small number of mutant heterozygotes still exist, which affect the development of bovine embryos by suppressing the expression of the ADIPOQ gene. In summary, the DD genotype can be used as a molecular marker to screen cows for superior traits and further improve herd fertility.” I recommend homogenizing with what was mentioned in line 273-276.

[Response 16] Thank you for your guidance and advice. To guarantee continuity and coherence in the conversation, we have reconciled the details you stated. The manuscript now keeps a coherent flow of information by rearranging these sections. Your advice was very helpful in raising the manuscript's overall standard of quality and clarity. We are grateful for your wise counsel.

[Comments 17] Conclusion :I recommend restructuring this section and being more specific about the findings mentioned in the results and described in “Discussion”.

[Response 17] Thank you for your guidance and advice. We have restructured the section as suggested and provided more specific details about the findings mentioned in the results, further elaborating on them in the "Discussion" section. These adjustments aim to enhance the clarity and coherence of the presentation of our research. Your input is greatly appreciated in refining the organization and focus of our manuscript.

[Comments 18] Line 303-304.- I recommend eliminating “The samples used were ovarian tissue (n=2111) and blood (n=108)”.

[Response 18] Thank you for your suggestion to streamline the content. We have removed the mention of "The samples used were ovarian tissue (n=2111) and blood (n=108)" as per your recommendation.

Reviewer 3 Report

Comments and Suggestions for Authors

Simple summary: looks good, no comments

Abstract: In line 35,36; authors mentioned “…study offer a theoretical foundation”, I would recommend to rephrase this part since you are presenting your own experimental data not reviewing others work and creating possible theoretical representation.

Introduction: In line 64, “tissueand” a space is missing, and same in line 91, “wasto”.

In figure 1, if authors mentioned which ones are follicle, corpus luteum, and albicans respectively. It will be better for readers to understand.

Material and methods: in section 2.2, processing tissues in saline water and weighing them and then stored for RNA, are you sure that you are not compromising RNA quality here?

Results: In figure 2C, it will be good for readers if authors mention beside the sequence figures which one is what (II or DD). Other than this rest of results were presented well.

Discussion: It is good that authors mentioned with importance the adverse effect of over-enhance production. This is really nice and necessary.

In line 281 authors mentioned ADIPOQ polymorphism can cause polycystic ovary syndrome in human. Can this predominant duplication of ADIPOQ gene in Chinese Holstein cow population also cause that? What is authors opinion in this context?

Conclusion: looks good, no changes necessary.

Author Response

[Comments 1] Abstract: In line 35,36; authors mentioned “…study offer a theoretical foundation”, I would recommend to rephrase this part since you are presenting your own experimental data not reviewing others work and creating possible theoretical representation.

[Response 1] Thank you for your guidance on the abstract. We appreciate your observation regarding the phrase "study offer a theoretical foundation" in lines 35-36 of the abstract. In response to your suggestion, we will rephrase this section to better reflect our experimental data and avoid implying theoretical representations beyond the scope of our study. Thank you for highlighting this point for clarification.

[Comments 2] Introduction: In line 64, “tissueand” a space is missing, and same in line 91, “wasto”.

[Response 2] Thank you for bringing this to my attention. I have made the requested change. The missing space in "tissueand" and "wasto" has been corrected. Your attention to detail is greatly appreciated.

[Comments 3] In figure 1, if authors mentioned which ones are follicle, corpus luteum, and albicans respectively. It will be better for readers to understand.

[Response 3] Thank you for your insights regarding Figure 1 in our manuscript. We value your suggestion to enhance the clarity and comprehensibility of the figure by explicitly identifying the components representing follicles, corpus luteum, and leukocytes. In light of your recommendation, we have refined Figure 1 by incorporating distinct labels that highlight the various structures depicted, including follicles, corpus luteum, and leukocytes. Your feedback is invaluable in refining the quality and accessibility of our research presentation. Thank you for your constructive input.

[Comments 4] Material and methods: in section 2.2, processing tissues in saline water and weighing them and then stored for RNA, are you sure that you are not compromising RNA quality here?

[Response 4] Thank you for your question, we use clean sterile saline solution to preserve the ovarian tissue and transport it to the laboratory within two hours, and we operate in a cryogenic environment and in a clean bench to ensure the quality of RNA as much as possible.

[Comments 5] Results: In figure 2C, it will be good for readers if authors mention beside the sequence figures which one is what (II or DD). Other than this rest of results were presented well.

[Response 5] Thanks for the suggestion, we have made changes to figure 2.

[Comments 6] In line 281 authors mentioned ADIPOQ polymorphism can cause polycystic ovary syndrome in human. Can this predominant duplication of ADIPOQ gene in Chinese Holstein cow population also cause that? What is authors opinion in this context?

[Response 6] Thank you for your attention. In our study, we mentioned that the polymorphism of the ADIPOQ gene may lead to human polycystic ovary syndrome. As of right now, no research has found a link between bovine polycystic ovarian syndrome and the polymorphism of the ADIPOQ gene. Our research merely established the polymorphism's association with ovarian traits; more testing is necessary to ascertain the polymorphism's result in polycystic ovary syndrome.